# The Mediating Effect of Workplace Spirituality on the Relation between Job Stress and Job Satisfaction of Cancer Survivors Returning to Work

**DOI:** 10.3390/ijerph16193510

**Published:** 2019-09-20

**Authors:** Ju-Hyun Jin, Eun-Ju Lee

**Affiliations:** 1College of Nursing, Research Institute of Nursing Science, Keimyung University, Daegu 42601, Korea; dominicajin@hanmail.net; 2College of Nursing, Keimyung University, Daegu 42601, Korea

**Keywords:** quality of working life, cancer survivors, workplace spirituality, job stress, job satisfaction

## Abstract

This study aimed to investigate the mediating effect of workplace spirituality in the relation between job stress and job satisfaction as well as the level of job stress, job satisfaction, and workplace spirituality of cancer survivors returning to work. A total of 126 cancer survivors who returned to work more than six months prior to the research participated in this study. Participants were recruited through snowball sampling; they were visiting the outpatient clinic at two general hospitals located in a metropolitan city and their clinical stage was stage 0 or stage 1. The collected data were analyzed using SPSS 22.0. Job stress, workplace spirituality, and job satisfaction had a negative correlation, whereas workplace spirituality and job satisfaction had a positive correlation. The Sobel test was performed to verify the significance of the mediating effect size of workplace adaptation, the results confirmed a partial mediating effect of workplace spirituality on the relation between job stress and job satisfaction (Z = –4.72, *p* < 0.001). This study confirmed the mediating effect of workplace spirituality in the relation between job stress and job satisfaction. A systematic program needs to be developed to enhance workplace spirituality, a spiritual approach, to relieve job stress and increase job satisfaction.

## 1. Introduction

According to 2016 data, the number of cancer patients in Korea was 1.74 million people, making up 3.4% of the total population; one person out of 29 was either currently receiving cancer treatment or a cancer survivor. Moreover, the cancer incidence in Korea was 269 out of 100,000 people, lower than the OECD (Organization for Economic Cooperation and Development) average of 300.3 people; meanwhile, the five-year relative survival rate was 70.6%, similar to or higher than that in the US (69.2%), Canada (60%), and Japan (62.1%). These outcomes are the result of the government-led cancer prevention and control policies and the advancement of cancer treatment technologies [1]. The rise in the number of cancer survivors who are leading their lives while managing cancer as a chronic disease means that a long-term approach is needed by shifting from the short-term cancer patient management in terms of prolonging life to the management of the quality of life. In this regard, the importance of managing the quality of life of cancer survivors, by helping them return to work and keep their jobs, has been emphasized. For cancer survivors, returning to work means returning to their previous everyday lives, which they could not enjoy anymore after being diagnosed with cancer and going through cancer treatment, as well as the recovery of interpersonal relationships [2]. Further, it is expected that working cancer survivors, who spend most of their time at work like many other healthy people, can achieve self-realization and enhance their self-esteem and job satisfaction, which can contribute to improving their overall quality of life [3]. Despite these positive aspects of returning to work, however, the number of cancer survivors who actually return to work makes up 30.5% of the total cancer patients, which is at a lower level compared with the average rate of returning to work for overseas patients at 63.5% [4].

This difference can be considered as coming from the differences in culture, social environment and systems, and personal and social perceptions between Korea and other countries [5]. The low rate of returning to work among Korean cancer survivors can be attributed to the influence of negative experiences by cancer survivors, including physical and mental health problems, stress from re-adapting to work and new duties, prejudices and discrimination of the organization and other members of the organization toward cancer survivors, and anxiety for the recurrence and metastasis of cancer [6]. Indeed, the job stress of cancer survivors returning to work is higher than that of healthy workers [3]. Such a higher level of job stress is caused by the adverse effects of cancer treatment, such as fatigue, pain, depression, and anxiety. The job stress of cancer survivors has been confirmed to be one of the important factors influencing the quality of working life, which refers to the subjective satisfaction at work [3], and it seems that the job stress of cancer survivors has a negative impact on job satisfaction: the job stress of workers with disabilities like cancer, which can be seen as a chronic disease, tends to have a negative correlation with their job satisfaction [7]. Indeed, a study on workers has shown that the higher the job stress, the lower the job satisfaction [8,9,10].

Moreover, the job stress of cancer survivors also has a negative correlation with workplace spirituality [3], which refers to a mental state under which an individual pursues and realizes the purpose and meaning of life through work while having a sense of community within the organization [11,12]. Previous studies have revealed that the workplace spirituality of cancer survivors exceeds the intermediate level, which is a higher level compared with the workplace spirituality level of nurses [3]; cancer survivors’ return to work is an important factor influencing their return to everyday life, recovery of confidence and self-esteem, and momentum for a mature life [2,13,14].

Meanwhile, in the study of workers without cancer, workplace spirituality is observed to have a positive correlation with job satisfaction [15,16,17], and is a factor influencing the quality of work life (QWL) of cancer survivors, which refers to the satisfaction level at work [3]. From these results, it can be expected that workplace spirituality can have a positive impact on the job satisfaction of cancer survivors returning to work. Although any mediating effect of workplace spirituality has not been discovered in the relation between job stress and job satisfaction of cancer survivors, workplace spirituality has been shown to have a partial mediating effect in the relation between career adaptability and job satisfaction of workers without cancer [18] and a moderating effect in the relation between excessive workload, which can cause job stress, and job satisfaction [19].

By comprehensively considering the results of the previous studies mentioned above, it can be estimated that the job stress of cancer survivors has a correlation with workplace spirituality and job satisfaction, and there exists a certain route among these three variables.

For cancer, treatment type and prognosis vary depending on the clinical stage [20], but even patients in the early clinical stages experience various difficulties, in terms of physical, emotional, psychological, interpersonal relationship, and social aspects [21], because cancer survivors in the cancer expansion stage after five years from the diagnosis of cancer are affected by anxiety for recurrence and metastasis [20].

Therefore, this study aimed to identify the mediating effect of workplace spirituality in the relation between job stress and job satisfaction among cancer survivors in stage 0 or stage 1, many of whom return to work [22], and who had finished the acute stage treatment. Moreover, there have not been sufficient studies on the effects of workplace spirituality, which alleviates job stress and job satisfaction of cancer survivors returning to work. In this regard, this study was conducted to provide the basic data necessary to search for the methods for spiritual approach and application to workers, as a means to relieve the job stress and enhance the job satisfaction of cancer survivors.

The purpose of this study was to investigate whether the relation between job stress and job satisfaction is mediated by workplace spirituality. Therefore, the hypotheses were: (1) job stress has a negative and significant effect on workplace spirituality, (2) job stress has a negative and significant effect on job satisfaction, and (3) workplace spirituality has a mediating effect on the relation between job stress and job satisfaction.

## 2. Methods

### 2.1. Study Design

This study employed a cross-sectional survey to identify the mediating effect of workplace spirituality in the relation between the job stress and job satisfaction of cancer survivors returning to work. Figure 1 presents the conceptual framework of the relationship between variables to identify the mediating effect of workplace spirituality in the relation between the job stress and job satisfaction of cancer survivors returning to work.

Few studies related to job stress, workplace spirituality, and job satisfaction among cancer survivors exist. In this study, the concept of job stress is considered to be unique. Specific occupational stressors in Korean employees with cancer based on organizational culture include hard work, fit to the organization, etc. [23,24]. According to previous studies, job stress was related to workplace spirituality and job satisfaction [8,9,10].

Meanwhile, the concept of workplace spirituality among cancer survivors includes the employees experiencing a sense of meaning at work, calling to the job, empathy with colleagues, etc., at the workplace, which positively contributes to job satisfaction and organization outcomes. Besides workplace spirituality, the concept of engagement also focuses on the spirit at workplace. Although the interest in these two concepts is continuously increasing, both the concepts have been studied independently. However, recently, workplace spirituality has emerged as significant for meaningfulness at work, engagement maintenance, and generalization [25].

Job stress and workplace spirituality are associated with job satisfaction, which includes the job itself, salary, promotion opportunities, supervision, and coworkers.

With regard to the relation between job stress and job satisfaction, although no studies have focused on cancer survivors returning to work, research has revealed a negative correlation between job stress and job satisfaction of workers with disabilities [7], and that the job stress of cancer survivors has a negative impact on the QWL, which refers to the subjective satisfaction at the workplace [3]. In addition, a negative correlation has been observed in nurses who showed a high level of job stress [8,9]. In a different report, the higher the job stress of those working at dental medical institutions, the lower their job satisfaction [10]. Meanwhile, workplace spirituality and job satisfaction have a positive correlation in the studies on people who were not cancer patients [15,16,17]. It has been confirmed that a higher level of workplace spirituality relates to higher job satisfaction in hospital nurses [16]. Meanwhile, in cancer survivors returning to work, workplace spirituality has shown a positive correlation with the QWL [3].

Moreover, although the mediating effect of workplace spirituality in the relation between job stress and job satisfaction of cancer survivors has not been confirmed, research on workers who were not cancer patients has shown that workplace spirituality has a partial mediating effect in the relation between career adaptability, which is the cause of job stress, and job satisfaction [18].

As such, based on the fact that job stress affects workplace spirituality, and workplace spirituality affects job satisfaction, the conceptual framework in the present study, is that workplace spirituality serves as a mediating factor in the relation between job stress and job satisfaction.

### 2.2. Sample and Setting

This study was conducted at a general hospital located in a metropolitan city in South Korea. The selection criteria for the participants were as follows: cancer survivors belonging to cancer stage 0 or stage 1 who had returned to work for six months after acute cancer treatment, such as operation and chemotherapy. Self-employed persons were excluded owing to the characteristics of the questionnaire items of study. The researchers obtained approval for data collection by explaining the purpose of the study to the hospital department in advance. The researchers visited the outpatient wards and directly collected data. To protect the rights of the study participants, those who understood the purpose and contents of this study and then agreed in writing to participate in the study were asked to complete the questionnaire. The participants were previously informed that they had the right to withdraw their participation at any time and were promised that collected data would be processed anonymously and would not be used for any purposes other than for those of this study.

The number of participants required for this study was verified as the number of samples for regression analysis using G*power 3.1 software program (Heinrich Hein University, Dusseldorf, Germany). If a medium effect size required for regression analysis is 0.15 with a power (1-*β* error probability) of 0.80 and significance level of *α* = 0.05, at least 123 participants are required. Therefore, considering a dropout rate of 10%, this study recruited a total of 136 participants; 126 completed questionnaire responses, after the exclusion of completed questionnaires with low reliability owing to incomplete information, were used as a valid sample.

### 2.3. Data Collection

Data were collected from the cancer survivors of one general hospital’s cancer center located in a metropolitan city; snowball sampling was used in parallel to account for the difficulties in selecting participants. This study was conducted after obtaining the cooperation of the nursing directors of the nursing department of two hospitals following relevant hospital procedures. The researcher obtained prior permission from the hospitals for data collection.

For protecting the privacy of the participants, the questionnaires were directly distributed to the outpatient wards by the researchers. The participants were asked to complete a consent form and a questionnaire and then completed questionnaires were directly collected by the researchers. It was stated in the questionnaire that completing the questionnaire takes 20–30 min, that the collected data were anonymized and would not be used for any purpose other than for those of this study, and that the participants could withdraw their participation at any time. A modest incentive was offered as a reward for participation.

### 2.4. Instruments

Demographic and clinical variables were obtained: sex, age, marital status, number of children, education level, occupation sector, managerial position, and years in the job; and cancer diagnosis, cancer stage, cancer treatment, current treatment, and periods of sick leave. Three instruments were used as well: job stress, job satisfaction, and workplace spirituality.

#### 2.4.1. Job Stress

The Korean Occupational Stress Scale (KOSS) Short Form, developed by Chang et al. [23] for general employees was used to measure the level of job stress among cancer survivors returning to work. KOSS was based on the Job Content Questionnaire (JCQ), Effort Reward Imbalance (ERI), the Generic Job Stress Questionnaire in NIOSH (National Institute of Occupational Safety and Health) and the Occupational Stress Index (OSI), which are widely used in Korea to measure job stress [23]. The questionnaire was derived from large-scale qualitative research considering the organizational culture of Korea (NSDSOS Project: 2002–2004). The questionnaire comprises 24 items with a four-point Likert scale and eight subscales: physical environment (three items), job demand (eight items), insufficient job control (five items), interpersonal conflict (four items), job insecurity (six items), organizational system (seven items), lack of reward (six items), and occupational climate (four items). The internal consistency alpha ranged from 0.51 to 0.82. The reliability coefficient of the KOSS in Chang et al.’s study [23] was 0.82 and 0.81 in our study.

#### 2.4.2. Job Satisfaction

The Minnesota Job Satisfaction Questionnaire (MSQ) was developed by Weiss et al. [26] and translated by Park [27] for Korean employees. The Korea-MSQ Short version by Park was used to measure the job satisfaction of cancer survivors returning to work. This tool includes 20 items with a five-point Likert scale ranging from 1 (not at all) to 5 (very well); higher total scores indicate better workplace spirituality. The reliability coefficient of the questionnaire was estimated as 0.87 in Park’s study and 0.91 in our study.

#### 2.4.3. Workplace Spirituality

The Workplace Spirituality Index (WSI) was developed by Roh and Suh (2014) [11] for measuring the workplace spirituality of Korean employees. The WSI was derived from a comprehensive five-factor model of workplace spirituality suggested by Roh and Suh (2014) [11]. The previous studies focused on the theories of existentialism and humanism to deal with workplace spirituality in the scientific realm with an approach to compartmentalize workplace spirituality from religion [11]. The WSI included the following five subscales: inner life toward oneself (five items), calling toward one’s work (five items), empathy toward one’s colleagues (five items), community toward one’s organization (four items), and transcendence above and beyond one’s ego (four items). The reliability coefficient of the questionnaire in Roh and Suh’s study was 0.91 and 0.92 in our study. The WSI comprises 23 items scored on a seven-point Likert scale ranging from 1 (not at all) to 7 (always very well); higher total scores indicate better workplace spirituality.

### 2.5. Statistical Analysis

SPSS Statistics version 22.0 (SPSS, Chicago, IL, USA) was used to analyze the survey data. The general characteristics of the participants were analyzed by real number and percentage; the levels of job stress, workplace spirituality, and job satisfaction were analyzed by mean, standard deviation, minimum value, and maximum value. Differences in job stress, workplace spirituality, and job satisfaction according to the general characteristics of the participants were analyzed using *t*-test and ANOVA. Using SPSS Statistics, the reliability coefficients of the instruments were estimated by calculating Cronbach’s alpha values. The survey data were further analyzed using Pearson’s correlation to identify correlations between job satisfaction and the other variables. To verify the mediating effect of workplace spirituality, multiple regression analysis was conducted with the three-step procedure of Baron and Kenny [28], and the indirect effect was confirmed using the Sobel test [29].

## 3. Results

### 3.1. General Characteristics of Participants and Difference in Job Stress, Workplace Spirituality, and Job Satisfaction

In this study, the demographic and clinical characteristics of the 126 participants are provided in Table 1 below. Especially, the mean age was 49 ± 8.62 years of participants and the majority of participants 46 (36.5%) were those aged 41–50 years. With regard to work position, clerks numbered 54 (42.9%) and managers numbered 49 (38.9%).

There were significant differences in job stress according to age (*F* = 2.86, *p* = 0.027), work position (*F* = 5.61, *p* = 0.005), and number of cancer treatments (*F* = 3.85, *p* = 0.024). Participants in the age group below 30 experienced higher job stress than others; the clerk group experienced higher job stress compared with others. However, the Scheffé test showed no significant difference between the numbers of cancer treatment groups. Although mean differences existed, the insufficient sample size in the number of cancer treatment groups may have limited the interpretation of this finding.

There were significant differences in workplace spirituality according to age (*F* = 3.12 *p* = 0.017) and occupational sector (*F* = 3.77, *p* = 0.026). Participants in the age group above 60 years had a higher workplace spirituality score than others; health care and service and sales group experienced low workplace spirituality as compared with others.

There were significant differences in job satisfaction according to age (*F* = 3.14, *p* = 0.017) and work position (*F* = 0.81, *p* = 0.002). The manager group had a higher job satisfaction compared with others, and there was no significant difference between the work position groups. However, the Scheffé test showed no significant difference between the age groups. Although mean differences existed, the insufficient sample size in the number of cancer treatments may have limited the interpretation of this finding.

### 3.2. Mean of Job Stress, Workplace Spirituality, and Job Satisfaction

Table 2 shows that the job stress mean score of cancer survivors was 42.36 (0–100, low stress level). The mean score for workplace spiritualty was 5.09 of 7.0 (above medium level), and the mean score for job satisfaction was 3.59 of 5.0 (above medium).

### 3.3. Correlation between Job Stress, Workplace Spirituality, and Job Satisfaction

Table 3 shows the correlation matrix between job stress and workplace spirituality. These variables had a medium association (−0.46). Job stress and job satisfaction scored −0.69, indicating a strong association. The correlation among workplace spirituality and job satisfaction was 0.55, indicating a strong association as well.

### 3.4. Mediating Effects of Workplace Spirituality on Job Stress and Job Satisfaction

Baron and Kenny’s [28] procedure (1986) for multiple regression was performed to analyze the mediation effects (Table 4). Their method is an analysis strategy for testing mediation hypotheses. In this method for mediation, there are two paths to the dependent variable. The independent variable (job stress) must predict both the dependent variable (job satisfaction) as well as the mediator (workplace spirituality). Mediation is tested through three regressions: job stress predicting job satisfaction, job stress predicting workplace spirituality, and job stress and workplace spirituality predicting job satisfaction.

In the first step, regression analysis was performed to verify the effects of job stress, the independent variable, on workplace spirituality, the mediating variable; statistically significant (*β* = −0.38, *p* < 0.001) results were found. In the second step, the effects of job stress, the independent variable, on job satisfaction, the dependent variable, were found to be statistically significant (*β* = −0.65, *p* <0.001). In the third step, regression analysis was performed to verify the effects of the mediating variable on the dependent variable. The results showed that the effects of the mediating variable, workplace spirituality, were statistically significant when controlling for job stress, the independent variable (*β* = −0.54, *p* < 0.001). Additionally, the effects of workplace spirituality on job satisfaction were found to be statistically significant (*β* = 0.28, *p* < 0.001). Significant general characteristic variables such as age, occupational sector, and work position were considered in all analyses. The explanatory power of the model used in this study was 54.8%. Since the absolute β value of job stress, 0.54 was smaller than the β value of 0.65 found in the third step, the mediating effect of workplace spirituality was ascertained. Meanwhile, the results of the Sobel test [29,30] performed to verify the significance of the mediating effect size of workplace spirituality, confirmed a partial mediating effect of workplace spirituality on the relation between job stress and job satisfaction (Z = −4.72, *p* < 0.001). Finally, the hypotheses of this study were supported.

The results of verifying the mediating effects of workplace spirituality on the relation between job stress and job satisfaction are shown in Table 4 and Figure 2. To confirm the basic assumptions of the regression analysis, the scatterplot of the residuals was analyzed. The residuals were uniformly distributed around zero, indicating that the linearity and homoscedasticity of the model were satisfied. In addition, the results of a P-P plot test for regression standardized residuals showed that the residuals were close to a 45-degree straight line, satisfying the normality of errors. Variance inflation factor was calculated for two variables to verify multicollinearity. It was calculated to be 1.20, confirming that there was no problem of multicollinearity (when it exceeds 10). In the autocorrelation test, the Durbin–Watson statistic was 2.02, which is close to 2, confirming that there was no autocorrelation between the independent variables.

## 4. Discussion

In modern times, as the number of cancer survivors increases, the issue of the method and strategy to help them to return successfully to their everyday life and work has emerged [2]. Cancer survivors’ return to work can help them overcome their economic hardships as well as enhance their satisfaction with life [14,31]. However, an increasing number of cancer survivors fail to return to their previous workplace, and they instead end up finding a new job. Therefore, it is important to identify variables related with turnover and understand the correlations among those variables [32]. In particular, as cancer survivors are the people who overcame challenging treatment procedures, such as surgery and radiation treatment, spirituality helps them develop an integrated personality [33] and lead an active life through positive reinterpretation of their current life [34]. Therefore, as cancer survivors can reduce their job stress after returning to work and enhance job satisfaction through spirituality, this study intended to validate the mediating effect of spirituality in the relation between the job stress experienced by cancer survivors after they return to work and job satisfaction.

First, the job stress of cancer survivors was 42.36 points on a 100-point scale, or at a lower-than-medium level. This figure is slightly higher than the job stress score of 39.2 points reported in a study using the same instrument for ordinary workers [35], and it was at a similar level to the job stress score of 42.70 points reported in a study that measured the job stress of cancer survivors using the same instrument [3]. Thus, the job stress of cancer survivors is higher than that of ordinary people after returning to work. In the case of cancer survivors, they experience a higher level of job stress owing to fear of returning to work, fear of recurrence of cancer, and attitudes of co-workers toward cancer survivors [4,36]. However, as most of the studies on the job stress of cancer survivors have been conducted from fragmental perspectives, further research needs to be conducted on the specific roles and working environment that tend to relieve or exacerbate their job stress.

The workplace spirituality score of cancer survivors was 5.09 points in this study, which was higher than that of ordinary workers at 4.08 points [17] and 4.86 points [18]. Spirituality helps people discover the meaning of life from negative events and to overcome their situations through intuitive perceptions and insights [34]. Therefore, the workplace spirituality of cancer survivors, who attempt to overcome their situations by finding out the meaning of life and reinterpreting their experiences of a negative event, which is cancer, may be more enhanced compared with ordinary workers. In a conceptual analysis on cancer survivors’ return to work, Son and Lee [2] argued that returning to work is an important factor in shifting the thoughts of cancer survivors from cancer to other issues and is one of the healing processes. Consequently, workplace spirituality has its importance as a factor to help people bounce from negative events to a positive lifestyle. However, as some of the previous studies on the spirituality of cancer survivors have focused on religion [37,38], more studies need to be conducted from various perspectives to help cancer survivors discover the meaning of life.

In the case of job satisfaction, the score reported in the present study was lower than that of ordinary workers, at 3.59 points. The job satisfaction of cancer survivors is particularly affected by cancer stigma, including workplace situation after returning to work and attitudes of organization members; that is, the organizational culture [36]. This tendency can be seen as playing an important role in increasing job stress [3]. In the present study, job stress was a factor that directly or indirectly affected job satisfaction. Such a result is consistent with those on workers who were not cancer survivors [8,9]: job stress has a negative correlation with job satisfaction. However, although it is expected that the job stress of cancer survivors after returning to work will show different patterns from that of ordinary workers, a comparison could not be easily conducted as there are not many quantitative studies. In the qualitative study of Heo et al. [13] on breast cancer survivors’ return to work, cancer survivors have physical difficulties in performing their work and treatment in parallel, while also becoming emotionally sensitive, leading to increased job-related stress.

The workplace spirituality of cancer survivors had a mediating effect in the relation between job stress and job satisfaction. Most of the previous studies have revealed a negative correlation between job stress and job satisfaction [7,8] and a positive correlation between workplace spirituality and job satisfaction [16,18,19]. One study [17] reported that workplace spirituality has a 64.5% influence on job satisfaction. In our study, job stress and workplace spirituality had a 53.8% influence on job satisfaction, consistent with the results of previous studies. In addition, workplace spirituality was found to have a mediating effect between job stress and job satisfaction, but a comparison is impossible for this owing to the lack of comparable results in previous studies. As a kind of power to make people positively interpret the current situation and actively respond to it [33], workplace spirituality plays a role in helping people overcome job stress as well as enhancing job satisfaction. Thus, enhancing the workplace spirituality of cancer survivors returning to work is believed to play a mediating role in leading cancer survivors to overcome psychological difficulties, including stress, and reach a positive level of job satisfaction.

Overall, a strategy to lower job stress and enhance workplace spirituality is needed to increase the job satisfaction of cancer survivors returning to work. As an increasing number of cancer survivors are returning to work, such a strategy needs to be developed at the national level through the operation of programs to help cancer survivors to return and adapt to their workplace successfully. In particular, such programs need to include workplace spirituality enhancement programs so that cancer survivors returning to work can positively reinterpret and actively respond to stressful situations at their workplace, such as prejudice and difficulties in physical and emotional adaptation to work.

This study has its limitations. First, more than half of the participants were recruited through snowball sampling with consideration for the characteristics of cancer survivors returning to work; thus, it is not easy to generalize the results of this study. Second, this study could not identify various variables that influence the job satisfaction of cancer survivors returning to work other than job stress. In particular, workplace spirituality, which leads cancer survivors to a stable life through reinterpretation of the meaning of life, can contribute to enhancing job satisfaction by mediating or moderating other negative variables. Therefore, future research needs to be conducted from various perspectives, including comparative studies on job satisfaction of cancer survivors returning to work in various professions, analyses on influencing factors, and investigations on the negative variables mediated and moderated by the workplace spirituality of cancer survivors.

## 5. Conclusions

The study concludes the results as follows: Job stress among cancer survivors returning to work has a significantly negative effect on job satisfaction. It implies that if the cancer survivors’ job stress related to work environment including security, job demand, and fatigue are reduced, job satisfaction can be increased. Furthermore, it can be applied for care the employees with chronic disease at the workplace. Job stress also has a significant negative effect on workplace spirituality. It implies that if cancer survivors’ job stress is reduced, workplace spirituality which includes meaningfulness at work and self-realization through the work would increase. Furthermore, high workplace spirituality creates high organizational commitment and quality of working life at the workplace. Workplace spirituality has a significant positive effect on job satisfaction. This implies that spiritual factors support cancer survivors’ job satisfaction which, in turn, affects organizational productivity and management of the human resource in organizations. It can contribute to developing programs such as meditation and relaxation therapy at the workplace. Finally, this study presents the partial mediating effect of workplace spirituality between job stress and job satisfaction. This implies that job stress was the main effect to job satisfaction and workplace spirituality can positively influence job satisfaction among cancer survivors. Developing interventions for increasing job satisfaction among cancer survivors would contribute to their returning to work.

## Figures and Tables

**Figure 1 ijerph-16-03510-f001:**
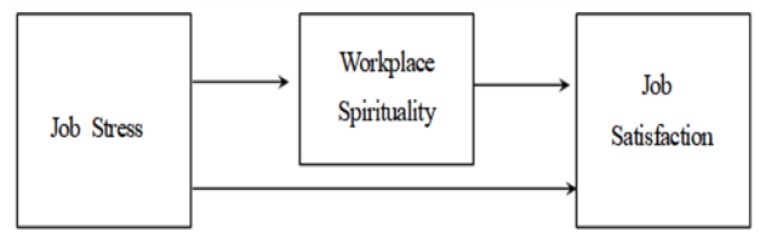
Conceptual framework.

**Figure 2 ijerph-16-03510-f002:**
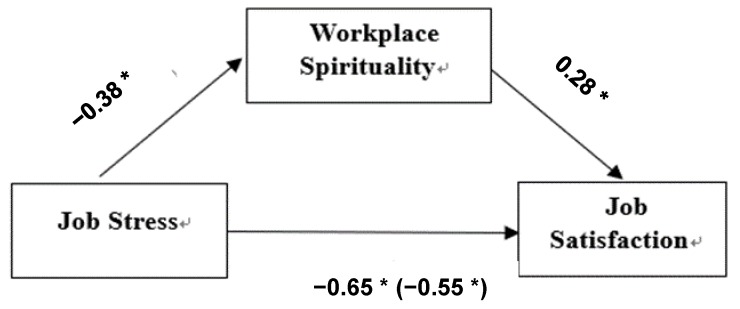
Mediating effect of workplace spirituality on the relation between job stress and job satisfaction. * *p* < 0.001, ** *p* < 0.05.

**Table 1 ijerph-16-03510-t001:** General characteristics of participants and differences in job stress, workplace spirituality, and job satisfaction (*N* = 126).

**Demographic Characteristics**			**Job Stress**		**Workplace Spirituality**		**Job Satisfaction**	
**Classification**		***N* (%)**	**M±SD**	***t* or *F* (*p*)**	**M ± SD**	***t* or *F* (*p*)**	**M ± SD**	***t* or *F* (*p*)**
Gender	Male	18 (14.3)	43.25 ± 8.18	0.39 (0.696)	4.90 ± 0.61	−1.18 (0.241)	3.49 ± 042	0.99 (0.324)
	Female	108 (85.7)	42.22 ± 10.66		5.13 ± 0.76		3.61 ± 0.49	
Age (years)	<30 years ^a^	4 (3.2)	49.21 ± 1.94	2.86 (0.027)	4.09 ± 0.24	3.12 (0.017)	3.28 ± 0.93	3.14 (0.017)
	31–40 years ^b^	15 (11.9)	41.40 ± 6.13	e < d < b < c < a	4.84 ± 0.98	a < b < c < d < e	3.47 ± 0.40	
	41–50 years ^c^	46 (36.5)	45.02 ± 9.91		5.07 ± 0.61		3.47 ± 0.42	
	51–60 years ^d^	51 (40.5)	35.00 ± 14.95		5.22 ± 0.76		3.70 ± 0.52	
	>60 years ^e^	10 (7.9)	42.37 ± 10.32		5.34 ± 0.59		3.88 ± 0.50	
Marital state	Single	21 (16.7)	38.76 ± 11.47	−1.77 (0.079)	5.22 ± 0.82	0.859 (0.392)	3.61 ± 0.53	0.23 (0.821)
	Married	105 (83.3)	43.09 ± 9.98		5.07 ± 0.73		3.59 ± 0.47	
Education level	High school	25 (19.8)	45.49 ± 9.51	1.70 (0.091)	4.90 ± 0.84	−1.15 (0.136)	3.48 ± 0.48	−1.18 (0.238)
	Higher than college	101 (80.2)	41.60 ± 10.41		5.14 ± 0.71		3.62 ± 0.48	
Number of Children	1	18 (14.3)	41.62 ± 9.46	0.43 (0.725)	4.98 ± 1.03	0.39 (0.756)	3.69 ± 0.57	2.09 (0.105)
	2	71 (56.3)	41.91 ± 10.18		5.14 ± 0.67		3.65 ± 0.45	
	≥3	15 (11.9)	45.16 ± 10.66		5.15 ± 0.67		3.41 ± 0.52	
	None	22 (17.5)	42.55 ± 11.55		5.00 ± 0.77		3.43 ± 0.43	
Occupation sector	Healthcare sector ^a^	49 (38.9)	44.01 ± 8.41	2.92 (0.057)	4.93 ± 0.69	3.77 (0.026)	3.56 ± 0.43	1.14 (0.323)
	Service and Sales ^b^	43 (34.1)	43.33 ± 11.5		5.07 ± 0.77	a, b < c	3.54 ± 0.48	
	Others ^c^	34 (27)	38.8 ± 10.68		5.60 ± 0.54		3.69 ± 0.53	
Work Periods (year)	<5 years	7 (5.6)	40.76 ± 8.21	0.98 (0.417)	4.85 ± 0.64	0.93 (0.449)	3.47 ± 0.36	1.94 (0.108)
	6–10 years	22 (17.5)	41.96 ± 11.05		4.86 ± 1.12		3.42 ± 0.53	
	11–15 years	12 (9.5)	39.98 ± 13.04		5.16 ± 0.65		3.58 ± 0.48	
	16–20 years	21 (16.7)	46.24 ± 7.62		5.16 ± 0.77		3.49 ± 0.44	
	>21 years	64 (50.8)	41.87 ± 10.46		5.17 ± 0.59		3.69 ± 0.48	
Working type	Fixed	30 (23.8)	43.51 ± 10.36	0.71 (0.480)	4.96 ± 0.81	−1.15 (0.252)	3.45 ± 0.45	−1.86 (0.064)
	Shift	96 (76.2)	42.00 ± 10.33		5.14 ± 0.72		3.63 ± 0.48	
Work position	Clerk ^a^	54 (42.9)	45.59 ± 9.50	5.61 (0.005)	4.92 ± 0.83	2.84 (0.621)	3.42 ± 0.46	1.81 (0.002)
	Manager ^b^	49 (38.9)	39.02 ± 10.83	b < c < a	5.20 ± 0.65		3.75 ± 0.47	a,c < b
	Others ^c^	23 (18.3)	41.94 ± 9.02		5.29 ± 0.64		3.61 ± 0.46	
**Clinical characteristics**		***N* (%)**	**Job Stress**		**Workplace Spirituality**		**Job Satisfaction**	
**Classification**			**M ± SD**	***t* or *F* (*p*)**	**M ± SD**	***t* or *F* (*p*)**	**M ± SD**	***t* or *F* (*p*)**
Cancer diagnosis	Breast ca.	42 (33.3)	43.44 ± 9.20	0.38 (0.767)	5.03 ± 0.79	0.29 (0.826)	3.49 ± 0.45	1.04 (0.376)
	Thyroid ca.	42 (33.3)	42.07 ± 10.77		5.18 ± 0.73		3.66 ± 0.48	
	Gastric ca	25 (19.8)	40.73 ± 11.46		5.05 ± 0.78		3.58 ± 0.56	
	Others	17 (13.6)	42.86 ± 10.67		5.11 ± 0.65		3.68 ± 0.41	
Onset periods from 2019	<1 year	10 (7.9)	40.20 ± 8.90	0.52 (0.672)	4.90 ± 1.07	1.21 (0.309)	3.71 ± 0.39	0.32 (0.810)
	1–5 years	65 (51.6)	42.08 ± 10.99		5.01 ± 0.74		3.56 ± 0.48	
	6–10 years	31 (24.6)	42.15 ± 11.04		5.19 ± 0.62		3.62 ± 0.50	
	11–20 years	20 (15.9)	44.74 ± 7.43		5.31 ± 0.73		3.59 ± 0.52	
Number of cancers treatment	1	64 (50.8)	40.66 ± 9.76	3.85 (0.024)	5.12 ± 0.78	0.29 (0.746)	3.65 ± 0.47	1.26 (0.287)
	2	37 (29.4)	46.24 ± 8.99		5.02 ± 0.67		3.49 ± 0.46	
	Others	25 (19.8)	41.02 ± 12.29		5.13 ± 0.78		3.56 ± 0.53	
Sick leave period (month)	≤3	83 (65.9)	42.02 ± 10.24	0.34 (0.795)	5.10 ± 0.78	1.17 (0.322)	3.60 ± 0.46	0.25 (0.859)
	4–6	19 (15.1)	41.94 ± 11.18		5.07 ± 0.72		3.62 ± 0.56	
	7–12	18 (14.3)	44.66 ± 9.46		4.95 ± 0.61		3.55 ± 0.45	
	≥13	6 (4.8)	41.73 ± 12.88		5.60 ± 0.54		3.45 ± 0.64	
Return to same workplace	Yes	112 (88.9)	42.09 ± 10.57	−0.85 (0.402)	5.09 ± 0.73	−0.23 (0.812)	3.60 ± 0.49	1.06 (0.288)
	No	14 (11.1)	44.59 ± 7.99		5.14 ± 0.89		3.46 ± 0.37	

POC: Scheffé test, a, b, c, d, e: marking of classification of variable.

**Table 2 ijerph-16-03510-t002:** Job stress, workplace spirituality, and job satisfaction of participants (*N* = 126).

Variables	Mean ± SD	Minimum Value	Maximum Value
Job stress	42.36 ± 10.3	5.95	67.8
Workplace spirituality	5.09 ± 0.74	2.39	6.78
Job satisfaction	3.59 ± 0.48	2.35	5.0

**Table 3 ijerph-16-03510-t003:** Correlation between job stress, workplace spirituality, and job satisfaction (*N* = 126).

Variables	Job Stress	Workplace Spirituality	Job Satisfaction
	*r* (*p*)	*r* (*p*)	*r* (*p*)
Job stress	1		
Workplace spirituality	−0.46 (<0.001)	1	
Job satisfaction	−0.69 (<0.001)	0.55 (<0.001)	1

**Table 4 ijerph-16-03510-t004:** Mediating effects of workplace spirituality on job stress and job satisfaction (*N* = 126).

Criterion Variables	Age	Occupation Sector	Work Position	Job Stress	Workplace Spirituality
	*β (p)*	*β (p)*	*β (p)*	*β (p)*	*β (p)*
**Step1** **Workplace** **spirituality**	0.17 **	0.12 (0.142)	0.09 (0.238)	−0.380 *	
			Adj R^2^	0.243	
			*F* (*p*)	11.02 *	
**Step2** **Job satisfaction**	0.15 **	−0.06 (0.348)	0.10 (0.132)	−0.652 *	
			Adj R^2^	0.492	
			*F*(*p*)	31.30 *	
**Step3** **Job satisfaction**	0.10 (0.107)	−0.09 (0.131)	0.07 (0.243)	−0.545 *	0.28 *
			Adj R^2^	0.548	
			*F*(*p*)	31.29^*^	

* *p* < 0.001, ** *p* < 0.05.

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
