# Peer review of "The Mediating Effect of Workplace Spirituality on the Relation between Job Stress and Job Satisfaction of Cancer Survivors Returning to Work"

_ijerph, 2019, doi:10.3390/ijerph16193510_

Round 1

Reviewer 1 Report

This is a good idea of research. It helps to design intervention programs for the professional reinsertion of cancer survivors.
The article is well organized and the hypotheses are checked using the appropriate methodology.
I would like to make two observations to the authors:
1.- What is the difference between the concept of workplace spirituality and engagment? It would be interesting to incorporate this commentary into the theoretical framework of the article.
2.- Why is a KOSS stress test used? Its reliability is a bit low (Alpha, 81). This aspect should be commented.

Author Response

1. What is the difference between the concept of workplace spirituality and engagement? It would be interesting to incorporate this commentary into the theoretical framework of the article. : It is added to 2.4.1. Study design, on page 3, below no. 121 Besides workplace spirituality, the concept of engagement also focuses on the spirit at workplace. Although the interest in these two concepts is continuously increasing, both the concepts have been studied independently..... 2. Why is a KOSS stress test used? Its reliability is a bit low (Alpha, 81). This aspect should be commented. → It was based on the followings; I did not mention on the paper. According to Nunnally(1978), Alpha, 70 is an acceptable reliability for affective domain, whereas Baumgartner and Jackson(1991) suggested Alpha, 80 for psychomotor measures. And Morrow and Jackson(1993) suggested the final adequacy of any particular reliability is a cognitive decision to be made by the test user. Therefore, KOSS’ reliability (Alpha .81) is acceptable for stress measurements in this study. Reference) • Nully, J. C. "Psychological Theory, 2nd." NY: McGraw-Hill (1978). • Baumgartner, Ted A., and Andrew S. Jackson. "Measurement for evaluation in physical education and exercise science. Dubuque: Wm. C." C. Publishers (1991). • Morrow Jr, James R., and Allen W. Jackson. "How “significant” is your reliability?" Research quarterly for exercise and sport 64.3 (1993): 352-355.

Reviewer 2 Report

In this cross-sectional study, the authors attempted to clarify the associations among job stress, job satisfaction, and workplace spirituality of cancer survivors who returned to work. Although the topic would draw relevant interests, the manuscript includes the following technical concerns to be addressed:

Major concerns

To assess job stress and workplace spirituality, the authors employed Korea-originated questionnaires that must not be familiar to the readers outside Korea. The relevant references only come from Korea. They need to explain the questionnaires in greater detail. See also the comments nos. 2 – 5. The authors must explain what kind of job stress factors can be assessed with the Korean Occupational Stress Scale (KOSS). There are some conceptual job stress models established worldwide, such as the Demand-Control-Support Model and the Effort-Reward Imbalance Model. What concept was the KOSS developed based on? What are the differences from other stress models? I cannot find what workplace spirituality is as it is written in the present style. In addition, the authors need to explain what concept the Workplace Spirituality Scale (WSS) was developed based on. The reliability of the KOSS and WSS was presented, however, their validity was not. I wonder whether there was conceptual overlap among job stress, job satisfaction, and workplace spirituality. The authors must explain whether these 3 factors are conceptually discriminated. Otherwise, the readers would doubt the common method variance/bias towards their associations found in the present study. Job satisfaction was significantly associated with age and work position and marginal-significantly associated with the number of children and working type. I wonder whether age, work position, the number of children, and working type were significantly associated with job stress and workplace spirituality. That is, I wonder that the subjects’ characteristics and working conditions could possibly work as confounders in the associations among job stress, job satisfaction, and workplace spirituality. The authors do not consider this possibility and did not adjust them in the analyses.

Minor concerns

(Lines 105 – 122) The authors explain the rationale for the study design. This description should go to the Introduction. The study hypotheses (lines 178 – 183) should go to the end of the Introduction. (Lines 218 – 219) Explanation on Baron and Kenny’s procedure should be done in the Methods in advance.

Author Response

Thank you.

Round 2

Reviewer 2 Report

N/A